# Association between metabolic risk factors and optic disc cupping identified by deep learning method

Jonghoon Shin[1,2], Min Seung Kang[1,2], Keunheung Park[3,4], Jong Soo Lee[3,4]*

1 Department of Ophthalmology, Pusan National University School of Medicine, Yangsan, South Korea,
2 Research Institute for Convergence of Biomedical Science and Technology, Pusan National University Yangsan Hospital, Yangsan, South Korea, 3 Department of Ophthalmology, Pusan National University College of Medicine, Busan, South Korea, 4 Medical Research Institute, Pusan National University Hospital, Busan, Korea

* jongsool@pusan.ac.kr

## Abstract

### Purpose

This study aims to investigate correlation between metabolic risk factors and optic disc cupping and the development of glaucoma.

### Methods

This study is a retrospective, cross-sectional study with over 20-year-old patients that underwent health screening examinations. Intraocular pressure (IOP), fundus photographs, Body Mass Index (BMI), waist circumference (WC), serum triglycerides, serum HDL cholesterol (HDL-C), serum LDL cholesterol (LDL-C), systolic blood pressure (BP), diastolic BP, and serum HbA1c were obtained to analyse correlation between metabolic risk factors and glaucoma. Eye with glaucomatous optic neuropathy(GON) was defined as having an optic disc with either vertical cup-to-disc ratio(VCDR) $\geq$ 0.7 or a VCDR difference $\geq$ 0.2 between the right and left eyes by measuring VCDR with deep learning approach.

### Results

The study comprised 15,585 subjects and 877 subjects were diagnosed as GON. In univariate analyses, age, BMI, systolic BP, diastolic BP, WC, triglyceride, LDL-C, HbA1c, and IOP were significantly and positively correlated with VCDR in the optic nerve head. In linear regression analysis as independent variables, stepwise multiple regression analyses revealed that age, BMI, systolic BP, HbA1c, and IOP showed positive correlation with VCDR. In multivariate logistic analyses of risk factors and GON, higher age (odds ratio [OR], 1.054; 95% confidence interval [CI], 1.046–1.063), male gender (OR, 0.730; 95% CI, 0.609–0.876), more obese (OR, 1.267; 95% CI, 1.065–1.507), and diabetes (OR, 1.575; 95% CI, 1.214–2.043) remained statistically significant correlation with GON.

**Data Availability Statement:** Data cannot be made publicly available because they contain identifying patient information. Data are available from the Pusan National University Yangsan Hospital Institutional Review Board (contact via +82-55-

360-3854) for researchers who meet the criteria for access to confidential data.

**Funding:** The authors received no specific funding for this work.

**Competing interests:** The authors have declared that no competing interests exist.

## Conclusions

Among the metabolic risk factors, obesity and diabetes as well as older age and male gender are risk factors of developing GON. The glaucoma screening examinations should be considered in the populations with these indicated risk factors.

## Introduction

Glaucoma is defined as a progressive optic neuropathy in which specific damage to the optic nerve and visual field (VF) defects occur. Although the most important risk factor for glaucoma is elevated intraocular pressure (IOP), the exact mechanisms underlying the anatomic and functional damage that occur in glaucoma remain unknown [1, 2]. Some risk for glaucoma may be due to systemic risk factors, as has been demonstrated by several previous studies on glaucoma pathogenesis [3, 4].

Metabolic syndrome (MS) is defined as comorbid obesity, hypertension, hyperglycaemia, and hyperlipidaemia. All of these conditions are themselves risk factors for cardiovascular disease and diabetes mellitus [5, 6], which can lead to ischemic vascular abnormalities and may influence glaucoma's pathogenesis [7, 8]. Despite this, previous studies have demonstrated conflicting results with regard to the association between metabolic syndrome and glaucoma [9, 10]. For instance, some large population-based studies have revealed that hypertension and diabetes mellitus are positively correlated with open angle glaucoma (OAG). However, other studies have reported no association between hypertension and OAG or diabetic mellitus and OAG. Previous studies have also demonstrated that higher BMI is associated with a lower risk of OAG [11, 12].

Because the prevalence of primary OAG with an IOP ≤ 21 mmHg in East Asia, including South Korea and Japan, is higher than that in most previous worldwide reports [13], health care centres have begun to use optic disc evaluation and IOP assessments for glaucoma screening in healthy adults. Then, this makes it possible to investigate the relationship between optic disc cupping and metabolic risk factors in data from the subjects visiting the health care centre.

The present study aims to clarify whether metabolic risk factors are associated with a higher or lower risk of OAG, an association that has remained unclear considering prior findings. Using data from the health care centre, we investigated whether metabolic risk factors are associated with optic disc cupping and the occurrence of glaucomatous optic neuropathy (GON).

## Materials and methods

This retrospective, cross-sectional study was performed in accordance with the tenets of the Declaration of Helsinki. The present study was approved by Institutional Review Board of Pusan National University Yangsan Hospital, South Korea. The requirement for informed consent was waived, as only data collected as part of routine health screening examinations was used.

### Subjects

Subjects who visited the health promotion centre at Pusan National University Yangsan Hospital from March 2009 to December 2018 and were 20 years or older were enrolled. All subjects underwent IOP measurements and fundus photography to screen for glaucoma. IOP was

measured with a non-contact tonometer (Canon T-2, Canon, Tokyo, Japan), and the mean value of IOP which was measured consecutively three times was calculated. Fundus photographs were taken using a digital non-mydriatic fundus camera (TRC-NW200; Topcon, Tokyo, Japan). In healthy subjects, the right eye of each subject was used for statistical analyses and in subjects with GON, data from eyes with glaucoma were obtained.

**Measurement of optic disc cupping and definition of glaucomatous optic neuropathy.** All fundus photographs were reviewed by deep learning system. We trained the state-of-the-art object-detection deep learning architecture, YOLO V3 [14]. The script codes for these architectures and darknet C source codes were directly downloaded from the homepage of the darknet and compiled in a Windows console application using Microsoft Visual Studio 2015 [15]. The hardware used included an Intel 8[th] generation central processing unit (CPU) (i5-8400, 2.81 GHz, 32 GB main memory) and an NVIDIA Titan Xp (12 GB; Santa Clara, CA, USA). Deep learning system was trained to find the location of the optic nerve head (ONH) and determine its vertical cup-to-disc ratio (VCDR) (S1 Fig). VCDR is a continuous number from 0 to 1 and, to label it, the number was binned by 0.1 starting from 0.3. A VCDR < 0.3 was labeled as '0.3' (which actually meant ≤0.3) because physicians also had greater disagreement when determining a VCDR < 0.3 [16]. Continuous measurements of the VCDR parameter were recorded, and eyes were classified with or without GON or using the following criteria: Eyes with GON were defined as having an optic disc with either VCDR ≥ 0.7 or a VCDR difference ≥ 0.2 between the right and left eyes.

**Definition of metabolic risk factors.** MS was defined according to the National Cholesterol Education Program Adult Treatment Panel III (NCEP ATP III), with Korean-specific guidelines for waist circumference. The components of MS according to these guidelines were waist circumference (WC), serum triglycerides, serum HDL-cholesterol, systolic and diastolic blood pressure (BP), and fasting glucose. In addition, given that obesity is a cause of metabolic abnormalities such as dyslipidaemia, hypertension, and diabetes mellitus, Body Mass Index (BMI) was also analysed in the present study.

The present study defined metabolic risk factors including BMI, WC, serum triglycerides, serum HDL cholesterol (HDL-C), serum LDL cholesterol (LDL-C), systolic BP, diastolic BP, and serum HbA1c. The Measurements of height, weight, and waist circumference (WC) were also taken for all subjects, and BMI was calculated as weight (kg) divided by height (m) squared. Blood samples were obtained and analysed for serum triglycerides, HDL-C, LDL-C, and HbA1c. Systolic and diastolic BP were measured with a brachial Riva-Rocci sphygmomanometer on the upper left arm with the patient in a seated position.

**Statistical analyses.** Data were collected and analysed separately for males and females. Statistical analyses were performed using SPSS version 20.0 (SPSS, Chicago, IL). Student's t-test and chi-square tests were used to compare demographic and clinical data between healthy eyes and those with GON. Pearson correlation coefficients were calculated to evaluate correlations between VCDR and age, BMI, systolic BP, diastolic PB, WC, serum triglyceride, HDL-C, LDL-C, HbA1c, and IOP. A multiple regression analysis with stepwise variable selection was conducted to assess the effects of metabolic risk factors on enlarging VCDR in the optic nerve head.

In risk factor analyses, the metabolic risk factors were categorized as follows. BMI was categorized into BMI less than 25 kg/m$^2$, and BMI of 25 kg/m$^2$ or more. BP was categorized into SBP / DBP < 130 / 80 mmHg, and SBP / DBP ≥ 130 / 80 mmHg. Participants were divided into 2 groups by WC: WC less than 90 cm for men and less than 85 cm for women, and WC of 90 cm or more for men and 85 cm or more for women. TG was categorized into TG less than 150 mg/dL, and TG of 150 mg/dL or more. HDL was categorized as follows: HDL of 40 mg/dL or more for men and 50 mg/dL or more for women, and HDL less than 40 mg/dL for men and

less than 50 mg/dL for women. Subjects were divided into 2 groups by HbA1c: HbA1c less than 6.5%, and HbA1c of 6.5% or more. For evaluating the potential risk factor for GON, univariate logistic regression analysis was conducted, and only variables with a *P* value of less than 0.10 were selected as independent variables in multivariate logistic regression model. Odds ratios with 95% CI values were calculated in all of the regression analyses. Statistical *P* value < 0.05 was considered statistically significant.

## Results

Among the 16,153 participants who underwent health screening examinations, 568 were excluded due to incomplete medical check-up records (n = 122) or poor fundus photography quality (n = 446). Of the remaining 15,585 subjects, 877 had GON defined with a VCDR ≥ 0.7 or a VCDR difference between the right and left eyes of ≥ 0.2 (Fig 1).

Table 1 compares the clinical characteristics of eyes with and without GON. Subjects with GON were more likely to be older (*P* < 0.001), male (*P* < 0.001), and had significantly higher BMI (*P* = 0.002), systolic BP (*P* < 0.001), diastolic BP (*P* < 0.001), WC (*P* < 0.001), HbA1c (*P* < 0.001), and lower LDL-C (*P* = 0.019) than those without GON.

Comparison of VCDR between eyes with GON and without GON area shown in Fig 2. In 877 eyes with GON, there was 673 eyes in VCDR 0.7, and 204 eyes in VCDR 0.8. 14708 eyes without GON were classified into 3891 eyes as VCDR 0.3, 2068 eyes as VCDR 0.4, 4619 eyes as VCDR 0.5, and 4130 eyes as VCDR 0.6. The mean VCDR value (0.723 ± 0.042) in eyes with GON were significantly higher than that (0.462 ± 0.115) in eyes without GON (*P* < 0.001) (Fig 2).

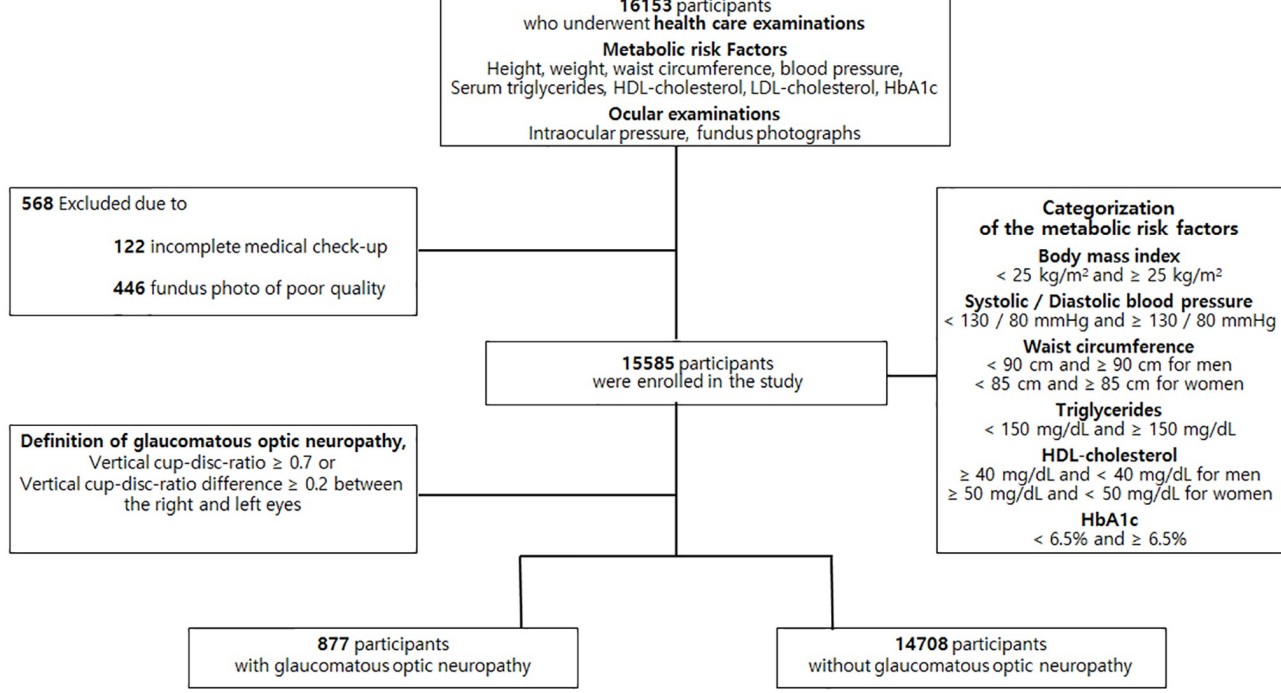

**Fig 1. Flow chart of eligible participants recruited from the Pusan National University Yangsan Hospital Health Study.** Among 16,153 participants underwent health care examinations, 15,585 participants were enrolled in study and 877 participants were diagnosed with glaucomatous optic neuropathy.

**Table 1. Subject characteristics based on the presence of glaucomatous optic neuropathy.**

| Characteristics | Eyes with GON | Eyes without GON | P value |
|---|---|---|---|
| | (n = 877) | (n = 14708) | |
| Age (years) | 54.74 ± 11.22 | 48.57 ± 10.84 | < 0.001[a] |
| Sex (male /female) | 584 / 293 | 8432 / 6276 | < 0.001[b] |
| Body mass index (kg/m$^2$) | 24.32 ± 3.39 | 23.95 ± 3.25 | 0.002[a] |
| Systolic blood pressure (mmHg) | 122.29 ± 14.71 | 119.31 ± 13.50 | < 0.001[a] |
| Diastolic blood pressure (mmHg) | 79.44 ± 10.45 | 77.98 ± 10.08 | < 0.001[a] |
| Waist circumference (cm) | 85.08 ± 8.88 | 83.67 ± 9.48 | < 0.001[a] |
| Triglyceride (mg/dL) | 125.90 ± 76.90 | 128.70 ± 92.79 | 0.302[a] |
| HDL cholesterol (mg/dL) | 53.66 ± 13.25 | 54.19 ± 12.84 | 0.249[a] |
| LDL cholesterol (mg/dL) | 124.98 ± 34.07 | 127.75 ± 34.11 | 0.019[a] |
| HbA1c (%) | 5.32 ± 1.65 | 5.03 ± 1.72 | < 0.001[a] |
| Intraocular pressure (mmHg) | 12.34 ± 3.25 | 12.21 ± 2.95 | 0.22[a] |

[a] By Student t test,

[b] By chi square test.

Correlation coefficients were calculated to evaluate the effects of age, BMI, systolic BP, diastolic BP, WC, triglyceride, HDL-C, LDL-C, HbA1c, and IOP on VCDR. Per univariate analyses, age, BMI, systolic BP, diastolic BP, WC, triglyceride, LDL-C, HbA1c, and IOP were significantly and positively correlated with VCDR (P < 0.001, 0.006, < 0.001, < 0.001, < 0.001, 0.007, < 0.001, < 0.001, < 0.001, respectively). HDL-C was negatively correlated with VCDR (P < 0.001) (Table 2).

When age, BMI, systolic BP, diastolic BP, WC, triglyceride, HDL-C, LDL-C, HbA1c, and IOP were entered into a linear regression analysis as independent variables, stepwise multiple regression analyses revealed that older age, higher BMI, higher systolic BP, higher HbA1c, and higher IOP were associated with larger VCDR in the optic nerve head (P < 0.001, 0.022, 0.037, 0.024, < 0.001, respectively) (Table 3) (Fig 3).

Univariate logistic analyses revealed that GON was significantly associated with older age, male gender, higher BMI, higher BP, and higher HbA1c (P < 0.001, < 0.001, 0.007, < 0.001, < 0.001, respectively). Those variables were evaluated in multivariable models with the covariates, using backward stepwise selection to eliminate those with P values > 0.05. Higher age, male gender, higher BMI, and higher HbA1c remained statistically significant via multivariable analyses (P < 0.001, < 0.001, 0.007, < 0.001, respectively) (Table 4). Moreover, the result of the predicting the GON has shown the 79.4% accuracy from multivariate logistic regression model.

## Discussion

In the present study, older age, higher IOP, higher BMI, higher systolic BP, and higher HbA1c were correlated with higher VCDR. In addition, of multiple metabolic risk factors, more obese and diabetes as well as older age and male gender were associated with an increased prevalence of GON. BMI had positively correlated with VCDR, and was also a risk factor for GON.

Given that BMI is a major indicator of obesity, previous studies have evaluated the association between BMI and glaucoma or increased IOP [8]. However, such prior work has revealed contradictory results on the relationship between BMI and glaucoma or BMI and IOP. Interestingly, some earlier studies demonstrated a negative association between BMI and glaucoma. This may result from impaired endothelium-dependent vasodilatation or a higher

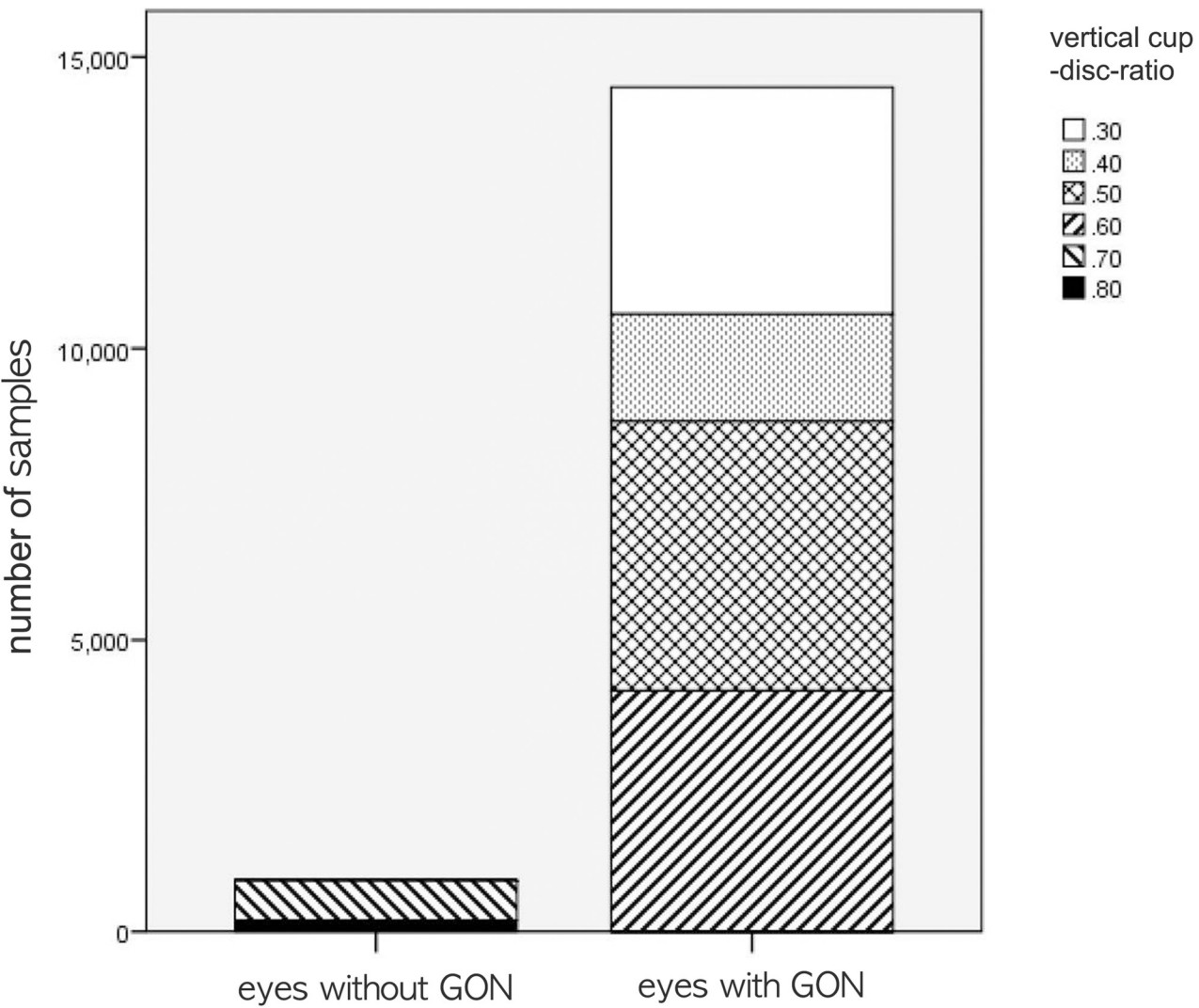

**Fig 2. Comparison of vertical cup disc ratio (VCDR) between eyes with glaucomatous optic neuropathy (GON) and without GON.** The independent t-test showed statistically significantly large VCDR in eyes with GON compared with eyes without GON ($P < 0.001$).

translaminar cribrosa pressure gradient induced by lower cerebrospinal fluid pressure, which is protective against glaucoma [17, 18]. Other studies reported that a higher BMI is a risk factor for glaucoma and further suggested that the positive association between BMI and glaucoma might be related to the presence of excess intraorbital fat tissue and increased resistance to outflow in the episcleral veins given increased blood viscosity with obesity [19, 20]. In agreement with the present study, a meta-analysis by Liu et al. revealed that adiposity (including BMI, WC, and waist-to-hip ratio) is associated with increased risk for elevated IOP and glaucoma [21]. The discrepancies in these results may be related to differences in their measurement methods, eligibility criteria, ethnicity, and participant composition. Additional studies are needed to illustrate more definitively, the relationship between obesity and glaucoma.

Higher HbA1c, an indicator of glycaemic control, is also positively correlated with VCDR and is a risk factor for glaucoma [3, 22]. Evidence from previous studies has supported the positive association found here between diabetes and glaucoma [3, 22]. Various contributing

**Table 2. Pearson correlation coefficients for vertical-cup-disc-ratio and age, metabolic risk factors, and intraocular pressure.**

|  | Vertical-cup-disc-ratio, P value[a] |
|---|---|
| Age | r = 0.182, P < 0.001 |
| Body mass index | r = 0.013, P = 0.006 |
| Systolic blood pressure | r = 0.086, P < 0.001 |
| Diastolic blood pressure | r = 0.074, P < 0.001 |
| Waist circumference | r = 0.047, P < 0.001 |
| Triglyceride | r = 0.028, P = 0.007 |
| HDL cholesterol | r = -0.025, P < 0.001 |
| LDL cholesterol | r = 0.017, P < 0.001 |
| HbA1c | r = 0.032, P < 0.001 |
| Intraocular pressure | r = 0.044, P < 0.001 |

[a] By Pearson correlation coefficients.

mechanisms have been proposed, including structural and functional abnormalities in the small vessels that feed the optic nerve, as well as increased susceptibility of retinal ganglion cells to apoptosis [23, 24]. In addition, Welinder et al. reported a positive association between HbA1c levels and glaucoma [25]. Zhao et al. further reported a significant relationship between HbA1c and glaucoma, suggesting that those with higher HbA1c's may be at greater glaucoma risk [23].

The present study also found a positive correlation between systolic BP and VCDR, despite the fact that elevated systolic BP was not a risk factor for glaucoma. The relationship between blood pressure and glaucoma remains controversial. A meta-analysis of the association between hypertension and open angle glaucoma revealed that systemic hypertension increases the risk of developing open angle glaucoma [26, 27]. Previous studies have also demonstrated that systemic hypertension may directly contribute to impaired microvasculature in the anterior optic nerve and increased IOP via overproduction or decreased outflow of aqueous humour due to abnormal autoregulation of the ciliary circulation [28, 29]. However, earlier studies also revealed that low blood pressure was a risk factor of developing glaucoma [30, 31], potentially leading to low perfusion pressure and subsequent glaucomatous changes in the optic nerve head (e.g., decreased rim area, increased cup area, and greater VCDR) [30, 32]. In addition, Jonas et al. reported that the neuroretinal rim area in healthy eyes without arterial hypertension was thicker than that in individuals with arterial hypertension, and that disc haemorrhage risk was also significantly different between glaucomatous eyes with and without arterial hypertension [18]. Though systolic BP may be positively associated with VCDR in the

**Table 3. Results of multivariate regression analyses of enlarged vertical cup-disc ratio.**

|  | Vertical-cup-disc-ratio | |
|---|---|---|
|  | B (95%) | P value[a] |
| Age | 0.002 (0.0019–0.0023) | < 0.001 |
| Body mass index | 0.001 (0.0003, 0.002) | 0.022 |
| Systolic blood pressure | 0.001 (0.0008, 0.0011) | 0.037 |
| HbA1c | 0.004 (0.002, 0.005) | 0.024 |
| Intraocular pressure | 0.002 (0.001, 0.003) | < 0.001 |

[a] By stepwise method.

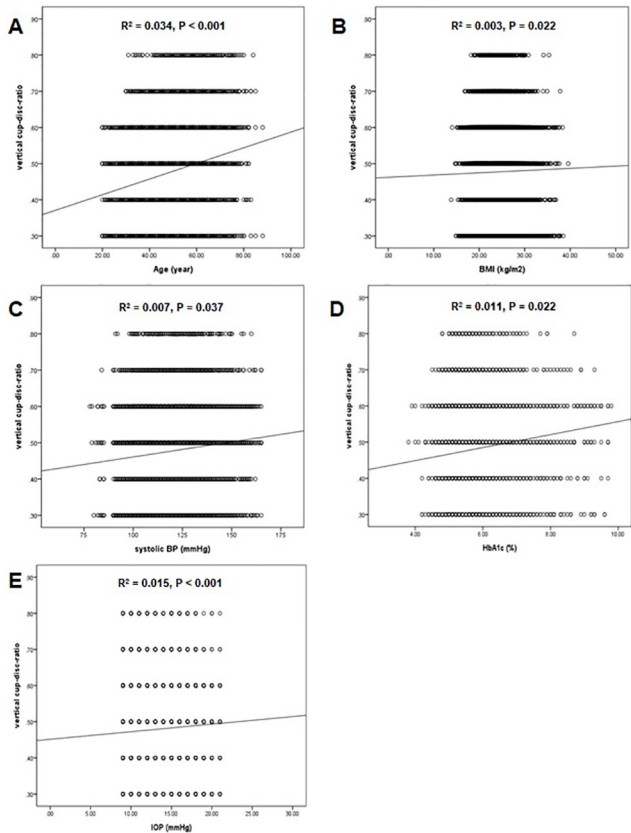

**Fig 3. Scatter plots showing the relationship between various factors and vertical cup-disc ratio (VCDR) in total subjects.** (A) Age and VCDR, (B) Body mass index and VCDR, (C) Systolic blood pressure and VCDR, (D) HbA1c and VCDR, and (E) intraocular pressure and VCDR. The dashed lines represent the 95% confidence intervals for the solid trend lines.

optic nerve head, various BP-related parameters including lower perfusion pressure, large BP fluctuations at night, systemic hypotension, antihypertensive treatments, and systemic cardio-vascular disease with hypertension may also contribute to glaucoma risk. Thus, further investigation is necessary to determine more comprehensively, the influence of BP changes on vascular insufficiency in glaucoma.

**Table 4. Logistic regression analyses of metabolic risk factors and their associations with glaucomatous optic neuropathy.**

| | Univariate | | Mutivariate[a] | |
|---|---|---|---|---|
| | OR (95% CI) | P | OR (95%) | P |
| Age | 1.053 (1.046, 1.060) | <0.001 | 1.054 (1.046, 1.063) | <0.001 |
| Gender (reference: Male) | 0.674 (0.584,0.779) | <0.001 | 0.708 (0.587, 0.854) | <0.001 |
| Body mass index (reference < 25 kg/m$^2$) | 1.214 (1.055, 1.397) | 0.007 | 1.267 (1.065, 1.507) | 0.007 |
| Blood pressure (SBP / DBP) (reference < 130 / 80 mmHg) | 1.376 (1.174, 1.613) | <0.001 | 1.068 (0.867, 1.317) | 0.534 |
| Waist circumference (reference < 90 cm or men, <85 for women) | 1.099 (0.957, 1.263) | 0.082 | 1.025 (0.856, 1.227) | 0.790 |
| Triglyceride (reference < 150 mg/dL) | 0.933 (0.799, 1.089) | 0.378 | 0.845 (0.694, 1.030) | 0.086 |
| HDL cholesterol (reference ≥ 40mg/dL for men, ≥ 50 mg/dL for women) | 0.997 (0.991, 1.002) | 0.916 | 0.981 (0.787, 1.224) | 0.867 |
| HbA1c (reference < 6.5%) | 2.274 (1.769, 2.922) | <0.001 | 1.606 (1.237, 2.086) | < 0.001 |
| Intraocular pressure | 1.016 (0.993, 1.039) | 0.099 | 1.016 (0.986, 1.048) | 0.299 |

[a] By backward stepwise logistic regression model.

In the present study, males had a higher prevalence of GON than females. In agreement with this, a large number of population-based studies using multivariate analysis approaches have indicated that males have a higher odds of developing glaucoma [2, 33–36]. This suggests that the low incidence of glaucoma in females may be related to the protective influence of endogenous oestrogen or exogenous hormone replacement following menopause [37, 38].

To the best of our knowledge, the present is the first report to evaluate continuously, the association between various metabolic risk factors and optic cup-to-disc ratio, a clinical marker of glaucomatous damage. Previous studies have simply evaluated anthropometric parameters (BP, glucose level, BMI etc) as risk factors for the prevalence of glaucoma. Measuring VCDR in the optic nerve head is essential for diagnosing glaucoma, and thus the continuous analysis of associated metabolic risk factors for increased VCDR in the optic nerve head is necessary to enhance our understanding of glaucoma risk factors. Despite its strengths, the present study does have several limitations, which warrant some discussion. First, it was not population-based but rather health centre-based, introducing a potential source of selection bias. Second, subjects did not undergo a complete glaucoma examination, including retinal nerve fibre layer photography, visual field examination, and optical coherence tomography, in the present study. We defined GON configuration as category 1, per the International Society of Geographical and Epidemiological Ophthalmology guidelines. However, optic nerve heads with GON cannot be diagnosed as glaucoma because category 1 requires visual field defects that correspond with glaucomatous damage. Third, we were unable to evaluate subjects for other ocular factors, such as axial length, refractive errors, or corneal thickness, which can affect optic disc measurements. Fourth, because the present study utilized a cross-sectional design, a causal relationship between glaucoma and associated metabolic risk factors cannot be established. Fifth, since the participants to undergo health screening examination enrolled in the health-care centre, the imbalance between normal subjects and study group was inevitably occurred in the present study. Finally, this study may have limitations from the inclusion of eyes with tilted discs, for which the optic disc analysis using deep learning system may have been erroneous. However, these ambiguous images were reviewed by an experienced ophthalmologist in an attempt to minimize any limitations.

In conclusion, our findings reveal that subjects with higher BMI, SBP, and HbA1c have greater VCDR in the optic nerve head. In a health centre-based Korean population, we also identified old age, male gender, obesity, and diabetes as significant GON risk factors. Given these findings, populations with the indicated risk factors should be targeted for glaucoma screening examinations, and population-based screening approaches should be implemented to reduce the under or misdiagnosis of glaucoma, which, if untreated, can result in severe outcomes including blindness.

## Supporting information

**S1 Fig. A diagram of deep learning architecture, YOLO V3 algorithm flow.**
(PNG)

## Author Contributions

**Conceptualization:** Jong Soo Lee.

**Data curation:** Jonghoon Shin.

**Formal analysis:** Jonghoon Shin, Min Seung Kang, Jong Soo Lee.

**Investigation:** Min Seung Kang, Jong Soo Lee.

**Methodology:** Keunheung Park.

**Resources:** Jonghoon Shin, Min Seung Kang, Keunheung Park.

**Software:** Keunheung Park.

**Supervision:** Jong Soo Lee.

**Writing – original draft:** Jonghoon Shin.

**Writing – review & editing:** Jong Soo Lee.

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
