## [Decision Letter · Decision Letter 0]

12 Jun 2020

PONE-D-20-07735

Association between metabolic risk factors and optic disc cupping identified by deep learning method

PLOS ONE

Dear Dr. Lee,

Thank you for submitting your manuscript to PLOS ONE. After careful consideration, we feel that it has merit but does not fully meet PLOS ONE’s publication criteria as it currently stands. Therefore, we invite you to submit a revised version of the manuscript that addresses the points raised during the review process.

The two reviewers have offered constructive criticisms that need to be addressed during revision. 

We look forward to receiving your revised manuscript.

Kind regards,

Sanjoy Bhattacharya

Academic Editor

PLOS ONE

Journal Requirements:

Reviewers' comments:

Reviewer's Responses to Questions

**Comments to the Author**

1. Is the manuscript technically sound, and do the data support the conclusions?

Reviewer #1: Yes

Reviewer #2: Partly

2. Has the statistical analysis been performed appropriately and rigorously? 

Reviewer #1: Yes

Reviewer #2: No

3. Have the authors made all data underlying the findings in their manuscript fully available?

Reviewer #1: Yes

Reviewer #2: No

4. Is the manuscript presented in an intelligible fashion and written in standard English?

Reviewer #1: No

Reviewer #2: Yes

5. Review Comments to the Author

Reviewer #1: There are only a few grammatical errors to correct.

In the abstract line 23 and 24, reword it to say 'with over 20 year-old patients THAT underwent health screening examinations'

Line 134: Should it read 'HDL was categorized as follows' with an 's' at the end of 'follow' ?

Line 136: divided into 2 'groups' instead of 'group'

Line 203: Add 'a' to 'BMI had positively correlated with VCDR, and was also a risk factor for GON.'

Reviewer #2: This paper did a great job in presenting a statistical analysis of correlations between various metabolic factors and Optic Disc Cupping, there are some major areas that need improvement. Some highlights: The discussion was clear, and the authors did a great job contextualizing their results with prior work; The paper was also organized nicely and there was a logical flow to the writing. Below are recommendations for revision:

The one major challenge with this paper was the obvious class imbalance. There was an almost 17:1 ratio of healthy subjects to GON subjects used in the statistical analysis. The authors should address this obvious class imbalance, and perhaps discuss the limitations of their models’ predictions given this fact. Other attempts to resolve this could include incorporating a focal loss cost function into their statistical methods.

To allow readers to better understand what the authors meant by the YOLO deep learning model, a system diagram detailing the chosen deep learning model architecture is required. The diagram should include abbreviated number of layers, feature maps, input and output. The trained model with weights should be included as a supplemental file or a link that can be accessed (I know models tend to be massive in size).

An additional system diagram detailing the flow of the entire process, from raw to final predictions would also be helpful for the reader to visualize the authors’ transition from raw clinical data to mathematical and clinical insights.

In regard to the statistical methods, there need to be clear mathematical equations, especially for the logistic/linear regressions. Describe reasons for using each statistical model. It was unclear if authors used a one tailed or two tailed t-test.

For results, the authors should not only include correlation coefficients, but also include the specificity and sensitivity (in the form of an F1 score perhaps). A confusion matrix would help readers visualize how univariate or multivariate logistic models performed in predicting GON.

There is much work to be done to improve this paper. Luckily, a lot of it is mostly computational, so I am confident the authors have the opportunity to improve upon the work.

6. PLOS authors have the option to publish the peer review history of their article (what does this mean?). If published, this will include your full peer review and any attached files.

Reviewer #1: Yes: Jada Morris

Reviewer #2: No

---

## [Author Response · Author response to Decision Letter 0]

12 Jul 2020

Dear Editor,

 We appreciate the reviewers for their efforts and the constructive comments. A point-by-point response to each of reviewers’ comments follows. The authors wish to thank the Editorial Board for their thoughtful consideration and recommendations, and hope that this revised manuscript now meets your requirements for publication.

<Reviewer #1>

There are only a few grammatical errors to correct.

In the abstract line 23 and 24, reword it to say 'with over 20 year-old patients THAT underwent health screening examinations'

Line 134: Should it read 'HDL was categorized as follows' with an 's' at the end of 'follow' ?

Line 136: divided into 2 'groups' instead of 'group'

Line 203: Add 'a' to 'BMI had positively correlated with VCDR, and was also a risk factor for GON.'

A) As your comments, we corrected grammatical errors in manuscript.

<Reviewer #2>

The one major challenge with this paper was the obvious class imbalance. There was an almost 17:1 ratio of healthy subjects to GON subjects used in the statistical analysis. The authors should address this obvious class imbalance, and perhaps discuss the limitations of their models’ predictions given this fact. Other attempts to resolve this could include incorporating a focal loss cost function into their statistical methods. 

A) We fully agreed with your comments. Unfortunately, the statistical consultant can’t suggest the definite solution about the obvious imbalance, so we described this imbalance between two groups in the limitation as following sentences; 

 “Fifth, since the participants to undergo health screening examination enrolled in the health-care centre, the imbalance between normal subjects and study group was inevitably occurred in the present study.”

To allow readers to better understand what the authors meant by the YOLO deep learning model, a system diagram detailing the chosen deep learning model architecture is required. The diagram should include abbreviated number of layers, feature maps, input and output. The trained model with weights should be included as a supplemental file or a link that can be accessed (I know models tend to be massive in size). 

A) As your comments, we attached the trained model as a supplemental file. After submitting this paper, the study about automatically measuring cup-to-disc ratio using deep-learning approach was published in Scientific Reports. (Park K, Kim J, Lee J. Automatic optic nerve head localization and cup-to-disc ratio detection using state-of-the-art deep-learning architectures. Scientific Reports. 2020.10:5025. doi.org/10.1038/s41598-020-62022-x). We can’t suggest more detailed algorithm of measuring CDR because of commercial availability.

An additional system diagram detailing the flow of the entire process, from raw to final predictions would also be helpful for the reader to visualize the authors’ transition from raw clinical data to mathematical and clinical insights. 

A) We fully agreed with your suggestion. Because the previous flow chart didn't include the detailed contents of metabolic risk factors, we added and revised the flow chart in Figure 1.

In regard to the statistical methods, there need to be clear mathematical equations, especially for the logistic/linear regressions. Describe reasons for using each statistical model. It was unclear if authors used a one tailed or two tailed t-test. 

A) As your comments, we clearly explained the mathematical equations for logistic regression model in the table 4 and manuscripts as following sentences.

“Univariate logistic analyses revealed that GON was significantly associated with older age, male gender, higher BMI, higher BP, and higher HbA1c (P < 0.001, < 0.001, 0.007, < 0.001, < 0.001, respectively). Those variables were evaluated in multivariable models with the covariates, using backward stepwise selection to eliminate those with P values > 0.05.”

For results, the authors should not only include correlation coefficients, but also include the specificity and sensitivity (in the form of an F1 score perhaps). A confusion matrix would help readers visualize how univariate or multivariate logistic models performed in predicting GON. 

A) We fully agreed with your comments. So, we added the specificity and sensitivity of the predicted probability as following sentence.

“ in logistic regression results, the predicted probability to discriminate between normal and GON showed 89.2 % specificity and 27.2 % sensitivity.”

---

## [Decision Letter · Decision Letter 1]

11 Aug 2020

PONE-D-20-07735R1

Association between metabolic risk factors and optic disc cupping identified by deep learning method

PLOS ONE

Dear Dr. Lee,

Thank you for submitting your manuscript to PLOS ONE. After careful consideration, we feel that it has merit but does not fully meet PLOS ONE’s publication criteria as it currently stands. Therefore, we invite you to submit a revised version of the manuscript that addresses the points raised during the review process.

A reviewer have raised a few criticisms. They need to be addressed to the extent possible by incorporating changes in the manuscript. A revised manuscript will be worthy of further consideration. 

We look forward to receiving your revised manuscript.

Kind regards,

Sanjoy Bhattacharya

Academic Editor

PLOS ONE

Reviewers' comments:

Reviewer's Responses to Questions

**Comments to the Author**

1. If the authors have adequately addressed your comments raised in a previous round of review and you feel that this manuscript is now acceptable for publication, you may indicate that here to bypass the “Comments to the Author” section, enter your conflict of interest statement in the “Confidential to Editor” section, and submit your "Accept" recommendation.

Reviewer #1: All comments have been addressed

Reviewer #2: All comments have been addressed

2. Is the manuscript technically sound, and do the data support the conclusions?

Reviewer #1: Yes

Reviewer #2: Yes

3. Has the statistical analysis been performed appropriately and rigorously? 

Reviewer #1: Yes

Reviewer #2: Yes

4. Have the authors made all data underlying the findings in their manuscript fully available?

Reviewer #1: Yes

Reviewer #2: (No Response)

5. Is the manuscript presented in an intelligible fashion and written in standard English?

Reviewer #1: Yes

Reviewer #2: (No Response)

6. Review Comments to the Author

Reviewer #1: (No Response)

Reviewer #2: Overall, I'm glad the researchers made significant improvements to the paper and it reads as more rigorous and technically sound.

Some additional comments:

1. Please also include the email of the IRB for data access (most researchers don't have access to international calls)

2. Figure 2 has "vertical cup disk ratio" on the y axis. Is this correct, or is it the number of samples?

3. Please explain the significance of a 89.2 % specificity and 27.2 % sensitivity. There is a stark contrast between the true positive and true negative rate; this needs to be discussed.

Thank you so much for the prior revisions; these are minor edits to an overall insightful study.

7. PLOS authors have the option to publish the peer review history of their article (what does this mean?). If published, this will include your full peer review and any attached files.

Reviewer #1: **Yes: **Jada Morris

Reviewer #2: No

---

## [Author Response · Author response to Decision Letter 1]

21 Aug 2020

Dear Editor,

 We appreciate the reviewers for their efforts and the constructive comments. A point-by-point response to each of reviewers’ comments follows. The authors wish to thank the Editorial Board for their thoughtful consideration and recommendations, and hope that this revised manuscript now meets your requirements for publication.

<Reviewer #2>

1. Please also include the email of the IRB for data access (most researchers don't have access to international calls)

A) The study protocol was approved by the Pusan National University Yangsan Hospital Institutional Review Board and you can access for data through below email address pnuyhirb@gmail.com.

2. Figure 2 has "vertical cup disk ratio" on the y axis. Is this correct, or is it the number of samples?

A) We make a mistake indexing on the y axis. As you would expect, index on the y axis was the number of samples. So, we changed the figure 2 as below.

3. Please explain the significance of a 89.2 % specificity and 27.2 % sensitivity. There is a stark contrast between the true positive and true negative rate; this needs to be discussed.

A) We fully agreed with your comments. As your comment in first revision, we previously analyzed the specificity and sensitivity for predicting GON by using multivariate logistic model, and described the 89.2% specificity and 27.2% sensitivity in the manuscript. Since cut-off value was set up as 0.5 automatically in SPSS program, we did not try calculating the various predicted probability as cut-off value was ranged from 0 to 1.0. According to the statistical consultant’s advices (Mi Sook Yoon PhD., msyun@pusan.ac.kr), we analyzed the all available parameters (accuracy, sensitivity, specificity, precision) showing the predicted probability in the range from 0.3 to 1.0 of cut-off value as below.

The low value of sensitivity in our study may be influenced by the parameters analyzed in logistic regression method, and could be improved as adding the useful diagnostic factors as the independent parameters in logistic regression analysis. In fact, the statistical results showed that the value of sensitivity was increased as the cut-off value was lower than 0.5, and then we should investigate the further evaluation including the ROC analysis.

In some studies of ophthalmogy (Sharma, et al. Post penetrating keratoplasty glaucoma: Cumulative effect of quantifiable risk factors. DOI: 10.4103/0301-4738.129790 / Lu et al. Comparison of Ocular Biomechanical Machine Learning Classifiers for Glaucoma Diagnosis. DOI: 10.1109/BIBM.2018.8621238), the goodness-of-fit of logistic regression models uses the accuracy of the classification table. Moreover, as your comments, there was outstanding gap between the sensitivity and specificity rate. Then, we and statistical consultant determined that the accuracy rate could be the appropriate parameter as probability for predicting GON.

So, we described these results as following sentences. (page 11, line 1)

“The result of the predicting the GON has shown the 79.4% accuracy from multivariate logistic regression model.”

---

## [Editor Report · Decision Letter 2]

31 Aug 2020

Association between metabolic risk factors and optic disc cupping identified by deep learning method

PONE-D-20-07735R2

Dear Dr. Lee,

We’re pleased to inform you that your manuscript has been judged scientifically suitable for publication and will be formally accepted for publication once it meets all outstanding technical requirements.

Kind regards,

Sanjoy Bhattacharya

Academic Editor

PLOS ONE
---

## [Editor Report · Acceptance letter]

3 Sep 2020

PONE-D-20-07735R2 

Association between metabolic risk factors and optic disc cupping identified by deep learning method 

Dear Dr. Lee:

I'm pleased to inform you that your manuscript has been deemed suitable for publication in PLOS ONE. Congratulations! Your manuscript is now with our production department. 

Kind regards, 

on behalf of

Dr. Sanjoy Bhattacharya 

Academic Editor

PLOS ONE